- 1 Quantifying the spatial-temporal patterns of land-atmosphere water, heat and 2 CO<sub>2</sub> flux exchange over the Tibetan Plateau from an observational perspective
- 3 Binbin Wang<sup>1,2,5,6,7</sup>, Yaoming Ma<sup>1,2,3,4,5,6,7\*</sup>, Zeyong Hu<sup>8</sup>, Weiqiang Ma<sup>1,2,4,5,6,7</sup>,
- 4 Xuelong Chen<sup>1,2,5,6,7</sup>, Cunbo Han<sup>1,2,5,6,7</sup>, Zhipeng Xie<sup>1,2,5,6,7</sup>, Yuyang Wang<sup>9</sup>, Maoshan
- 5 Li<sup>10</sup>, Bin Ma<sup>1,5,6,7</sup>, Xingdong Shi<sup>1,4,7</sup>, Weimo Li<sup>1,4</sup>, Zhengling Cai<sup>1,4</sup>
- 6 <sup>1</sup>State Key Laboratory of Tibetan Plateau Earth System, Resources and Environment (TPESRE),
- 7 Institute of Tibetan Plateau Research, Chinese Academy of Sciences, Beijing 100101, China;
- <sup>2</sup>College of Earth and Planetary Sciences, University of Chinese Academy of Sciences, Beijing
   100049, China;
- <sup>3</sup>College of Hydraulic & Environmental Engineering, China Three Gorges University, Yichang
   443002, China;
- 12 <sup>4</sup>College of Atmospheric Science, Lanzhou University, Lanzhou 730000, China;
- <sup>5</sup>National Observation and Research Station for Qomolongma Special Atmospheric Processes and
   Environmental Changes, Dingri 858200, China;
- <sup>6</sup>Kathmandu Center of Research and Education, Chinese Academy of Sciences, Beijing 100101,
   China;
- <sup>7</sup>China-Pakistan Joint Research Center on Earth Sciences, Chinese Academy of Sciences,
   Islamabad 45320, Pakistan;
- 19 <sup>8</sup>Key Laboratory of Land Surface Process and Climate Change in Cold and Arid Regions,
- Northwest Institute of Eco-Environment and Resources, Chinese Academy of Sciences, Lanzhou,
   730000, China;
- <sup>9</sup>College of Grassland Science and Technology, China Agricultural University, Beijing,
   100193, China;
- <sup>10</sup>School of Atmospheric Sciences, Chengdu University of Information Technology, Chengdu
   610225, China
- Corresponding authors: Binbin Wang (wangbinbin@@itpcas.ac.cn); Yaoming Ma
   (ymma@itpcas.ac.cn); Address: Building 3, Courtyard 16, Liucui Road, Chaoyang District, Beijing
   City, China. Zip code: 100101.

# 29 Abstract

30 Land-atmosphere (LA) interaction process, through the turbulent exchange of water, heat and CO2 31 flux, significantly influences regional micro-climates, local water cycles, energy budgets, and 32 ecosystem dynamics. The Tibetan Plateau (TP), characterized by vast extent, high elevation, strong 33 solar radiation and convection, as well as extreme weather fluctuations, has been under-explored due to 34 the scarcity of LA interaction stations, particularly in the western and northern regions. To address this 35 gap, this study introduces a newly constructed research and observation platform, which consists of 16 36 planetary boundary layer towers, spans diverse landscapes and covers dynamic meteorological 37 conditions, with average annual air temperature, wind speed and liquid precipitation ranging from -3.5 38 to 18.5 °C, 0.6 to 5.6 m s<sup>-1</sup>, and 43 mm to 2164 mm. Elevation correlates significantly with all 39 meteorological variables, highlighting a strong spatial heterogeneity distribution patterns of LA 40 coupling. The turbulent flux of water and heat show clear seasonal variations, with highest sensible 41 heat flux (SH) in April-May and largest latent heat flux (LE) in July-August. Further, most stations 42 report negative net ecosystem exchange (NEE) values, ranging from -3.2 g C m<sup>-2</sup> a<sup>-1</sup> to -174.3 g C m<sup>-2</sup> 43 a<sup>-1</sup>, and function as carbon sinks. However, Medog station, locating in the densely forested Yarlung 44 Zangbo valley, functions as a carbon source which is most probably related to the vegetation 45 destruction and human activities. LE is significantly and closely correlated with SH, NEE and 46 ecosystem respiration, indicating strong coupling between water, heat and carbon fluxes. Precipitation 47 as well as soil water content provide favorable moisture sources and show significance in the 48 water-carbon coupling process. The observation and research platform and the quality-controlled 49 high-temporal resolution data provide valuable in situ measurements for studying water-heat-carbon 50 interactions, validating numerical models and satellite algorithms, and offering ground truth for research on hydrological, meteorological, and ecological responses to global climate change. 51

52 **Key worlds**: Land-atmosphere turbulent flux, spatial-temporal variations, comprehensive 53 observation and research platform, Tibetan Plateau

# 54 **1 Introduction**

55 Land-atmosphere (LA) interaction, which governs the flux exchanges of energy, water, and CO2 56 between the Earth's surface and the atmosphere, are pivotal in shaping regional water cycles, climate 57 dynamics, and ecosystem changes [Gentine et al., 2019; Y Ma et al., 2023; Santanello et al., 2018; Seo 58 and Ha, 2022; Y Zhang et al., 2024]. The thermal contrasts between distinct landscapes—such as land 59 vs. water, mountain vs. valley, and ocean vs. land-drive both regional circulations, like lake-land and 60 mountain-valley breezes, and large-scale atmospheric motions, including monsoons [Gerken et al., 61 2014; Wu and Zhang, 1998; Wu et al., 2023]. These interactions are forced by distinct LA interaction 62 processes, which can further influence air pollution dispersion, atmospheric moisture transport, 63 regional convergence/divergence, redistribution of clouds and precipitation as well as ecosystem 64 evolution and carbon budget [Bei et al., 2018; Friedlingstein et al., 2022; Suni et al., 2015; Zhu et al., 65 2017]. For example, the intensified coupling of soil moisture and land surface temperature can 66 exacerbate droughts and heatwaves in northern East Asia (1980-2019) [Seo and Ha, 2022], where soil 67 moisture deficits reduce evapotranspiration (ET), amplifying heatwave conditions, particularly in areas 68 with sparse vegetation. Further, the LA water and heat coupling can even impact the intricate non-linear 69 feedback between ET and cloud water content, especially in transitional zones where uncertainties 70 remain under energy-limited and water-limited ET regimes [Y Zhang et al. [2024]. Under global 71 climate warming, the carbon absorption/release capacity through LA interactions by abrupt permafrost 72 thaw, plant uptake and ecosystem respiration have also been debated globally, especially over the data 73 scarce regions [Turetsky et al., 2020; Y Wang et al., 2023b; Wei et al., 2021]. Therefore, understanding 74 and quantifying the full spectrum of LA coupling through the in situ measurements of energy, water, 75 and  $CO_2$  flux is essential for comprehending the Earth system's response to climate change.

76 Understanding LA interactions through coordinated, multidisciplinary, and multi-scale 77 observations is crucial for addressing global challenges such as water resource management, land-use 78 planning, climate change, and ecosystem preservation. In this context, key global initiatives-such as 79 the First International Satellite Land Surface Climatology Project Field Experiment (United States) [P J 80 Sellers et al., 1992], the Hydrologic Atmospheric Pilot Experiment (France, Niger) [André et al., 1986; 81 Goutorbe et al., 1997], the Northern Hemisphere Climate Processes Land Surface Experiment (Sweden) 82 [Halldin et al., 1999], the Boreal Ecosystem-Atmosphere Study (Canada) [P Sellers et al., 1995], the 83 Inner Mongolia Semiarid Grassland Soil-Vegetation-Atmosphere Interaction and the Heihe River 84 Basin Field Experiment (China) [Liu et al., 2018; Lü et al., 1997]-have provided foundational 85 insights into LA interactions and have advanced parameterizations for climate models. Tibetan Plateau 86 (TP), the world's largest and highest plateau, plays a particularly critical role in the climate and ecology 87 dynamics. TP exerts significant influence on atmospheric processes, generating thermal disturbances 88 that affect circulation patterns, weather, and climate not only in China and East Asia but also 89 globally[Wu and Zhang, 1998; Ye and Wu, 1998]. For example, mesoscale system vortices and shear 90 lines created in the TP's atmospheric boundary layer can lead to extreme weather events, such as heavy 91 rain and storms, impacting both the plateau and surrounding regions [L Li et al., 2020; Xu and Chen, 92 2006]. Thus, LA coupling and dynamics are important for the formation and development of weather 93 systems. Over the past few decades, large-scale field activities and long-term observational 94 experiments-such as QXPMEX, TIPEX-I, TIPEX-II, TIPEX-III, JICA, GAME/Tibet, and 95 CAMP/Tibet-have greatly enhanced our understanding of land surface processes in the TP [Huang et

96 al., 2023; Y Ma et al., 2023], and these efforts have also helped refine climate model parameterization, 97 improving our ability to predict TP's climatic effects. However, the stations for measuring heat, water 98 and CO2 exchange are concentrated mostly in the east and still rarely distributed over the vast northern 99 and western regions, hindering our understanding on its spatial distribution and total amounts of heat, 100 water and CO<sub>2</sub> flux. Given the growing challenges posed by global climate change, accurately 101 measuring and modeling LA interaction processes is more critical than ever, and such efforts are 102 essential for predicting climate extremes, managing water resources, and supporting sustainable 103 ecology [Suni et al., 2015].

104 LA interaction station has decades of experiences in analyzing the dynamic LA interaction 105 parameters, the seasonal variations of turbulent flux and the driving forces behind them [Y Ma et al., 106 2005; Y Ma et al., 2018; Y Ma et al., 2023; Wang et al., 2011; Yang et al., 2008]. Meanwhile, 107 comprehensive LA interaction stations have been widely used to assess and to evaluate water and heat 108 turbulent fluxes derived from satellite algorithms and numerical simulations. The water-carbon coupled 109 biophysical model (PML-V2) has been used to generate a data set for ET and its components over the 110 period 1982-2016. Validation using eddy covariance (EC) measurements confirmed the reliability of 111 the results, yielding an average ET value of  $353 \pm 24$  mm yr<sup>-1</sup>. Simultaneously, based on EC 112 measurements, soil moisture, and soil texture data, [Ling Yuan et al., 2021] developed an improved ET 113 model and created a monthly ET data set covering the period from 1982 to 2018 [L. Yuan et al., 2024]. 114 Validation at nine EC stations demonstrated that the model outperforms previous studies, with an 115 average annual ET value of approximately  $346.5 \pm 13.2$  mm yr<sup>-1</sup>, which is highly consistent with the 116 results of [N Ma and Zhang, 2022]. However, factors influencing the inter-annual variations in ET 117 exhibit significant biases and uncertainties, especially for data-scarce western and northern regions. As 118 for carbon function, TP contains extensive permafrost and a variety of landscapes, including alpine 119 meadows, alpine steppes, alpine shrubs, alpine wetlands, forests, and alpine deserts, which have a 120 significant impact on the carbon sink/source function of the region, and shows important ecological and 121 environmental consequences. Recent studies indicate that most alpine meadows on the TP function as 122 carbon sinks, with values ranging from -430 to -12.5 g C m<sup>-2</sup> a<sup>-1</sup>, and some alpine steppe areas act as 123 weak carbon sources [Y Wang et al., 2023a; Wei et al., 2021]. Specifically, alpine grasslands exhibit a 124 weaker carbon sink function, with values ranging from -206.9 to -17.1 g C m<sup>-2</sup> a<sup>-1</sup> whilst shrub lands 125 show even lower carbon sink values, ranging from -89.5 to -40.7 g C m<sup>-2</sup> a<sup>-1</sup>. Marshes display 126 considerable variability in carbon fluxes, with values ranging from  $-187 \pm 29$  g C m<sup>-2</sup> a<sup>-1</sup> in Shenzha to 127 -478 g C m<sup>-2</sup> a<sup>-1</sup> in Haibei [*Qi et al., 2021; Zhao et al., 2005*]. Therefore, by synthesizing EC and 128 climate data from multiple sites across the TP, we can understand clearly the spatial patterns of water, 129 heat, CO<sub>2</sub> flux and identify the mechanisms that control them.

130 This study introduces a comprehensive observation and research platform for LA water, heat, and 131 CO<sub>2</sub> flux exchange over the TP (Figure S1 and Table S1) and provides a preliminary analysis of data 132 collected from 15 stations spanning more than 2 years recently, especially over the data-limited regions 133 of western and northern TP. The objectives are to address the following scientific questions: (1) What 134 are the characteristics of land-atmosphere water, heat, and CO2 flux exchange across different 135 landscapes of the TP? (2) What are the spatial and temporal distributions of water, heat, and CO<sub>2</sub> flux, 136 and what factors influence these variations? Specifically, the development of the observation platform, 137 instrument configuration, and standardized data processing methods introduced in the "Methods". The

138 preliminary results and discussions on the spatial-temporal variations of meteorological variables,

139 energy budget components, and CO<sub>2</sub> flux exchange are illustrated in the "Results and Discussion".

## 140 **2** The observation platform and methods

### 141 **2.1 Introduction of observation platform and instruments configuration**

142 The LA interaction process is essential in driving regional air motion and water cycles and 143 influencing weather and climate change. The EC system is widely used as a direct measure-or ground 144 truth-for estimating energy and material flux at the LA interface. Over time, long-term and 145 quasi-continuous EC networks have been established across various ecosystems and land covers 146 globally, including networks in North and South America (AmeriFlux and Fluxnet-Canada), Europe 147 (EuroFlux and CarboEurope), Asia (ChinaFlux and AsiaFlux), and Australia (OzFlux) [Baldocchi, 148 2014; Yu et al., 2024]. The TP has an area of 2.6 million km<sup>2</sup> and presents extreme environmental 149 conditions, such as high solar radiation, low temperatures, large diurnal temperature variations, limited 150 precipitation, nutrient-poor soils, shallow soil profiles, and short growing seasons. These factors 151 contribute to strong convection and intense weather variations. However, TP is also characterized as a 152 region lacking EC stations, particularly in the arid and semi-arid northern and western parts. To address 153 this gap, the Institute of Tibetan Plateau Research, Chinese Academy of Sciences (ITPCAS), 154 established six comprehensive and long-term LA interaction stations in remote and data-scarce regions 155 of TP gradually since 2004 [Y M Ma et al., 2008], including QOMOS (the Qomolangma Atmospheric and Environmental Observation and Research Station, CAS), NAMORS (the Nam Co Monitoring and 156 157 Research Station for Multisphere Interactions, CAS), SETORS (the Southeast Tibet Observation and 158 Research Station for the Alpine Environment, CAS), NADORS (the Ngari Desert Observation and 159 Research Station, CAS), MAWORS (the Muztagh Ata Westerly Observation and Research Station, 160 CAS), and Shuanghu. Since 2019, the 6 stations were upgraded with new sonic anemometer and gas analyzer sensors gradually (CSAT3 and LI7500DS, details in Table S1), enhancing the measurement 161 162 capabilities. The instrumentation and long-term data at 5 stations covering 2006-2021 can be found in 163 [*Y Ma et al.*, 2020] and [*Y Ma et al.*, 2024].

164 Furthermore, the Second Tibetan Plateau Expedition and Research Program (STEP) in 2019 has 165 expanded the network, adding 10 additional LA interaction stations, including Medog, Qamdo, 166 Mangkam, Mangai, Baingoin, Nyima, Jyirong, Burang, Coqen, Lhasa, especially in the remote western 167 and northern regions. The integrated eddy turbulence devices (IRGASON, Campbell; CSAT3 & 168 LI7500RS in Lhasa), capable of measuring high-frequency (10 Hz) quantities of sonic temperature, 169 water, CO<sub>2</sub>, and three-dimensional winds, have been used. This expansion resulted in the creation of 170 the Third Pole Environment Integrated Three-dimensional Observation and Research Platrom 171 (TPEITORP, observation platform for short hereafter) for measuring water, heat, and CO2 flux over the 172 TP [Y Ma et al., 2023]. All the stations have a 20 m planetary boundary layer (PBL) tower 173 measurements (in Lhasa the configuration height is 40 m), including traditional variables such as air 174 pressure, liquid precipitation, infrared land surface temperature, four-component radiation, soil 175 temperature and moisture at multi-layers, air temperature, air humidity and wind at 5 layers, as well as 176 turbulent flux of water, heat and CO<sub>2</sub>. The details of the instrument configuration, station locations, and 177 photographs are provided in Table 1 and Figure 1, respectively. In-situ measurements from these 178 stations, with updated systems, were utilized to analyze the seasonal and diurnal variations of water, 179 heat, and CO<sub>2</sub> exchange, as well as the energy budget and carbon source/sink dynamics across

- 180 contrasting ecosystems and climates. A detailed illustration of the observational environments at 16
- 181 stations are as follows.