# Peer review of "Quantifying the spatial-temporal patterns of land-atmosphere water, heat and 2 CO2 flux exchange over the Tibetan Plateau from an observational perspective"

_Earth System Science Data, 2025_

## Referee Comment (RC2)

Review of Wang and Ma et al., manuscript number essd-2025-195 "Quantifying the spatial-temporal patterns of land-atmosphere water, heat and CO2 flux exchange over the Tibetan Plateau from an observational perspective"

This study presents a new dataset that observes variables related to meteorological conditions, as well as water, energy, and carbon fluxes at the Tibetan Plateau. The 16 measurement stations are distributed across a broad gradient of climate and landscape types. The authors present the temporal dynamics of key measured variables and analyze the relationship between energy fluxes and environmental variables.

The efforts in collecting the data and making it public are appreciated. However, the manuscript needs revision to be clearer and more informative, as outlined below:

**General comments:**

- 1. The introduction to this topic is currently very general. It would be beneficial if the authors could suggest more specific topics to highlight the niche of this dataset. What knowledge can be gained using this dataset, particularly distributed at the TP? What are the potential research gaps in understanding the fluxes across the observed land cover types? What is the motivation for measuring fluxes at these sites? The potential contribution to the scientific community from this dataset needs to be more clearly elaborated.
- 2. The inconsistency in result presentation makes the structure of the paper not very straightforward. For instance, the analysis between energy fluxes and climate variables across dry and wet stations provides interesting results. However, it might be beneficial to clarify why this analysis is particularly necessary for energy fluxes instead of carbon fluxes, and why different temporal resolutions (diurnal, daily, monthly) are shown for different variables as main figures.

**Specific comments:**

- L108-116: These results are very specific, even presented with numbers. Are these
  results specific to TP? There are many land surface models calibrated with the EC
  flux measurements. Is there a particular reason to highlight this PML-V2 model here?
  And why is the predictability of ET particularly introduced here, instead of, for
  instance, GPP?
- 2. L128: In addition to the spatial patterns, temporal variations can also be highlighted.
- 3. L142-152: These sentences seem to belong to the introduction section.
- 4. L172: Are the turbulent fluxes measured on the same 20 m PBL tower? How would the tall tower affect the EC measurements?
- 5. L179: Does the "CO2 exchange" have the same meaning as "carbon source/sink dynamics"?
- 6. What are the precise vegetation types? What is the forest species at the Medog site? It's helpful to provide detailed information for further research.
- 7. Sometimes, it is difficult to capture information from the text descriptions. For instance, the paragraph L184-196 can be more informative if summarized into a table.
- 8. L239-242: Are most of the data gaps related to precipitation?

- 9. How to define "abnormal values" (L246)? What do you mean by "carefully checked and removed" (L249)? And how is the gap-filling performed (L261)? Please provide clear data processing guidelines.
- 10. Why is NEE calculated, not measured?
- 11. L325: Isn't precipitation also one of the meteorological variables in section 3.1?
- 12. Figure 4: why are the data aggregated at monthly values? It can be informative to show the data for each day of year as well (e.g., Figure 2 in Paulus et al., 2022)
- 13. L409-414: Can you give info about which reason(s) applies for which site(s)?
- 14. L435: How is  $\Delta T$  and  $\Delta E$  calculated? Why is  $\Delta T$  analysed only with SH instead of LE?
- 15. Figure 5: Are these correlation coefficients all statistically significant?
- 16. L456: Why is it doubled? This sentence is misleading.
- 17. L526-530: It is not clear what exactly leads the site Medog to be a carbon source. Is it contributed by disturbances or climate extremes? What causes the strong seasonal variation of GPP at this site (Figure 6)?

**Technical comments:**

- 1. L137: Incomplete sentence: "...are introduced..."
- 2. L229: analyze -> analyzing;
- 3. L231: analyzing -> analyze
- 4. L369: Do you mean energy flux?

**References:**

Paulus, S. J., El-Madany, T. S., Orth, R., Hildebrandt, A., Wutzler, T., Carrara, A., Moreno, G., Perez-Priego, O., Kolle, O., Reichstein, M., and Migliavacca, M.: Resolving seasonal and diel dynamics of non-rainfall water inputs in a Mediterranean ecosystem using lysimeters, Hydrol. Earth Syst. Sci., 26, 6263–6287, https://doi.org/10.5194/hess-26-6263-2022, 2022.

---

## Author Comment (AC1)

We would like to thank the anonymous reviewer for the useful comments and suggestions. We have taken advantage of the advices for improving the manuscript. For an easier comprehension, the reviewer's comments are included in our rebuttal in black typeface. Our replies appear in blue typeface. Line numbers appeared in the rebuttal are from the revised version, with changes highlighted with red color in the revised document.

**Reviewer 1**

This manuscript presents a valuable and comprehensive dataset on turbulent water, heat, and CO2 fluxes across the Tibetan Plateau, based on measurements from 16 eddy covariance stations. The authors highlight notable spatial heterogeneity in meteorological conditions and land-atmosphere exchange processes. Given the scarcity of long-term, ground-based observations in this region, the dataset is relevant and will be useful for model validation and understanding flux dynamics on the Plateau.

That said, there are several areas that could be improved to strengthen the manuscript:

**Responses:**

We sincerely thank the reviewer for the positive evaluation of our dataset and for recognizing its scientific value and relevance to the study of land-atmosphere exchanges over the Tibetan Plateau. We appreciate the reviewer's thoughtful comments and constructive suggestions for improving the manuscript and we have revised the manuscript following the valuable comments and suggestions.

While the REddyProc is referenced for flux partitioning, key methodological details are missing — particularly the u\* threshold filtering, gap-filling strategy, and quality control protocols. These are essential for transparency and reproducibility.

**Responses:**

We thank the reviewer for the valuable comments. In the revised manuscript, we have expanded the description of the flux partitioning procedure conducted with REddyProc to include details on the u\* threshold determination, gap-filling approach, and quality control procedures.

The u\* threshold was estimated using the bootstrapping approach implemented in REddyProc, following the standard procedure described by Wutzler et al. (2018). Thresholds were determined separately for each year, and the mean value was applied to exclude nighttime periods with insufficient turbulence. Flux gap-filling was conducted using the marginal distribution sampling method within REddyProc, based on relationships with radiation, air temperature, and vapor pressure deficit, using a 7–14 day moving window. Quality control involved removing data points with known

sensor malfunctions, spikes, or physically implausible fluxes, and applying the flagging scheme. Low-quality or filtered data were excluded prior to gap-filling and partitioning.

These details have been added to the revised section 2.2 "Methods for data processing and analyzing" (Lines 293–300) to improve clarity, reproducibility, and alignment with community standards.

Phrases like "significantly correlated" are used frequently, but without supporting statistical values (e.g., p-values, confidence intervals, or correlation coefficients). This weakens the interpretation of the results.

**Responses:**

We appreciate the reviewer's valuable comments, and we agree that including the statistical metrics strengthens the interpretation of our results. In the revised manuscript, we have added the corresponding correlation coefficients (r, the Pearson correlation coefficient) and p-values (p<0.05 or p<0.01) for all reported relationships previously described as "significant." Where relevant, the correlation is statistically significant at the 95% or 99% confidence level. For some words like "significant" or significantly, they have been revised as "remarkable", "pronounced", "considerable", "substantially", "markedly", and others.

These additions now ensure that each statement of statistical significance is quantitatively supported. The revised results and figure captions have been updated accordingly, i.e. Lines 326, 332-334, 337, 363, 364, 391, 398, 432, 472, 481, 513, 517, 527, 535, 566, 603, 618, Figure 5, Figure S1, Figure S2, Table S7.

The manuscript classifies stations into "wet" and "dry," but it's unclear how this was defined. Some objective metric like aridity index, annual precipitation thresholds, or soil moisture levels would help readers understand the basis of this classification.

**Responses:**

We thank the reviewer for pointing out the need for greater clarity regarding the classification of sites as "wet" and "dry." Generaly, the classification of climatic zones can be determined by using the aridity index (AI), defined as the ratio of total precipitation to total potential evapotranspiration over a given period; the formula for AI is given as: AI = Prec/ET, where Prec denotes the annual mean precipitation, and ET denotes the annual mean potential evapotranspiration. AI > 0.65 indicates humid regions, 0.5-0.65 semi-humid regions, 0.2-0.5 semi-arid regions, 0.03-0.2 arid regions, and < 0.03 hyper-arid regions (Programme, 1997; Programme, 2022). Some in-depth analysis has been conducted following this concept.

In this manuscript, the wet and dry stations are roughly categoried following the status of precipitation and soil moisture, i.e. wet stations have a higher precipitation (> 400 mm) and larger soil moisture contents (SM10>10%). In contrast, the dry stations have lower precipitation or lower soil moisture contents. We have clarified these information in Line 475-479 of the revised manuscript.

**References:**

Programme, U.N.E., 1997. World Atlas of Desertification: Second Edition.

Zomer, R.J., Xu, J., Trabucco, A., 2022. Version 3 of the Global Aridity Index and Potential Evapotranspiration Database. Sci Data 9, 409. https://doi.org/10.1038/s41597-022-01493-1

The discussion occasionally loses focus, with hydrology, meteorology, and carbon cycle topics mixed together without strong connections. Consider restructuring or summarizing the key takeaways more clearly.

**Responses:**

We thank the reviewer for this thoughtful comment. We agree that strengthening the connections among meteorological, hydrological, and carbon fluxes components can improve readability and focus. In this manuscript, the Results and Discussion are presented as a combined section, where each subsection (3.1 - 3.4) focuses on a distinct aspect of the land - atmosphere exchange process—meteorological variables, precipitation and soil moisture, energy fluxes, and carbon fluxes (NEE, GPP, and Re).

We have retained this integrated structure to maintain consistency with the Earth System Science Data format, in which the main purpose is to describe, evaluate, and provide access to the dataset rather than to conduct extensive process-based interpretation. To improve clarity, we have strengthened the transitions between subsections to better link meteorological, hydrological, and carbon processes. We have also added a brief summary paragraph at the end of each subsection. Check line 368-372, line 416-420, 520-524, 596-601 of the revised manuscript. These revisions help clarify the focus of each section and make the takeaway message more clearly.

There are a number of minor grammatical issues and typos throughout. I've flagged several below but recommend a full proofreading pass before submission.

**Responses:**

Thanks for the careful corrections and comments. The entire manuscript has undergone a comprehensive proofreading and language revision to correct minor

grammatical issues and typographical errors. These edits have improved readability and consistency throughout the whole manuscript.

Line-by-Line Comments

Title: The title suggests strong emphasis on temporal patterns, but the manuscript only presents seasonal analysis, there's little interannual variation discussed. You might consider rephrasing.

**Responses:**

We appreciate the reviewer's helpful comment. We agree that the original title implied a broader temporal scope than what is actually analyzed. As our observations cover approximately two years, the study primarily focuses on spatial, seasonal, and diurnal variations of land—atmosphere exchanges of water, heat, and CO2, rather than long-term interannual changes.

To better reflect the dataset and analyses presented, we have revised the title to emphasize both the spatial distribution and the short-term temporal variations of the observed fluxes. The revised title reads: "Quantifying the spatial-seasonal patterns of land-atmosphere water, heat and CO2 flux exchange over the Tibetan Plateau from an observational perspective"

This version accurately captures the scope of the dataset and the analyses presented, highlighting the spatial coverage of the observation stations and the focus on short-term temporal variability derived from in-situ measurements.

Line 43: Probably should be "located" rather than "locating."

**Responses:**

Thanks for the correction. We have revised the abstract and the whole sentence to avoid the grammar error.

Line 52: Key worlds?

**Responses:**

Thanks for the correction. We have corrected "worlds" as "words" following the comment.

Line 56: Use "is pivotal" instead of "are pivotal," since "interaction" is singular.

**Responses:**

Thanks for the comments. We have revised the whole sentence following the comment. Check line 78-79 of the revised manuscript.

Lines 94–95: It's not clear what these names refer to, consider clarifying.

**Responses:**

Thanks for pointing out this ambiguity. We have revised the manuscript to clarify what each name refers to upon its first mention. Check line 115-119 of the revised manuscript.

Table 2 vs Line 408: Table 2 includes "EBC" — did you mean "EBR"? Also, line 408 references Table 1 for EBR, but the data appears in Table 2. Please check consistency.

**Responses:**

We thank the reviewer for catching this inconsistency carefully. "EBR" is correct—the term "EBC" in Table 2 was a typographical error and should indeed read "EBR", referring to the energy balance ratio. We have corrected this label in the table and ensured that all occurrences of EBC/EBR throughout the manuscript are now consistent. Check Table 2 of the revised manuscript.

Additionally, we have corrected the table reference in the text: the values for EBR are presented in Table 2, not Table 1 as previously indicated. Check line 460 of the revised manuscript.

Lines 123–125: Are these numbers from your own results? If so, clarify. If not, cite the sources.

**Responses:**

Thanks for the comments. The values are from the supplymental material of Table S1 of Wei, et al. 2021. We have clarified their origin in the revised manuscript. Check line 145 of the revised manuscript.

Line 131: "CO2 flux exchange" is redundant, consider "CO2 flux".

**Responses:**

Thanks for the correction. We have revised the sentence to void redundancy following the comment. Check line 151-152 of the revised manuscript.

Line 170: "Platrom"?

**Responses:**

Thanks for the correction. We have corrected the word to "platform". Check line 195 of the revised manuscript.

Line 231: "to analyzing" should be "to analyze."

**Responses:**

Thanks for the correction. We have revised the word following the comment. Check line 259 of the revised manuscript.

Line 254: When referencing "WPL correction," please explain what WPL stands for (Webb-Pearman-Leuning)?.

**Responses:**

We thank the reviewer for the helpful comments. Yes, WPL stands for Webb-Pearman-Leuning, referring to the density correction proposed by Webb, Pearman, and Leuning (1980). We have revised the manuscript to explicitly define the term WPL correction upon its first mention.

Specifically, WPL correction accounts for the effects of air density fluctuations caused by heat and water vapor transfer on the measured CO2 and H2O fluxes, ensuring accurate computation of mass and energy exchange between the surface and atmosphere. Check line 283 of the revised manuscript.

Line 499 / Figure 6: Figure 6 only shows 12 stations, but the manuscript states 15 were used. Please clarify which stations were excluded and why.

**Responses:**

Thanks for the comments. We appreciate the reviewer's careful observation. In the revised manuscript, we have clarified that three stations—QOMOS, NAMORS, and Baingoin—were excluded from the CO2 fluxes analysis shown in Figure 6.

These sites were removed because the diurnal variations of net ecosystem exchange (NEE) were found to be physically unrealistic or inconsistent with expected flux patterns: At QOMOS and NAMORS, the diurnal cycles of NEE exhibited inverted daytime and nighttime patterns, suggesting potential sensor issues. At Baingoin, abnormal nighttime CO2 release was observed, which is likely associated with local biomass (cow manure) burning for heating, as reported by field technicians during the observation period.

Given these anomalies, data from the three stations were not included in the regional synthesis and subsequent analysis to ensure data quality and physical consistency. The manuscript has been updated accordingly in line 305-310 of the revised manuscript.

Line 513: "carbon absorption primarily occur" → should be "occurs."

**Responses:**

Thanks for the correction. We have revised the word following the comment. Check line 571 of the revised manuscript.

---

## Author Comment (AC2)

We would like to thank the anonymous reviewer for the useful comments and suggestions. We have taken advantage of the advices for improving the manuscript. For an easier comprehension, the reviewers' comments are included in our rebuttal in black typeface. Our replies appear in blue typeface. Line numbers appeared in the rebuttal are from the revised version, with changes highlighted with red color in the revised document.

**Reviewer 2**

Review of Wang and Ma et al., manuscript number essd-2025-195 "Quantifying the spatial-temporal patterns of land-atmosphere water, heat and CO2 flux exchange over the Tibetan Plateau from an observational perspective"

This study presents a new dataset that observes variables related to meteorological conditions, as well as water, energy, and carbon fluxes at the Tibetan Plateau. The 16 measurement stations are distributed across a broad gradient of climate and landscape types. The authors present the temporal dynamics of key measured variables and analyze the relationship between energy fluxes and environmental variables.

The efforts in collecting the data and making it public are appreciated. However, the manuscript needs revision to be clearer and more informative, as outlined below:

**Responses:**

We sincerely thank the reviewer for the careful evaluation and constructive feedback. We appreciate the recognition of our efforts to establish and share this multi-site flux dataset across the Tibetan Plateau. We have addressed all of the reviewer's specific comments in detail and improved the manuscript's clarity and coherence.

**General comments:**

1. The introduction to this topic is currently very general. It would be beneficial if the authors could suggest more specific topics to highlight the niche of this dataset. What knowledge can be gained using this dataset, particularly distributed at the TP? What are the potential research gaps in understanding the fluxes across the observed land cover types? What is the motivation for measuring fluxes at these sites? The potential contribution to the scientific community from this dataset needs to be more clearly elaborated.

**Responses:**

We thank the reviewer for these valuable suggestions. We agree that the "Introduction" should more explicitly emphasize the scientific motivation, knowledge gaps, and unique contributions of the dataset. Accordingly, we have substantially

revised and expanded the Introduction to clearly show the significance of our work. Specifically, we have added the following elements:

- (1) Motivation and rationale We now highlight the crucial role of the Tibetan Plateau (TP) as a climate-sensitive and data-scarce region where quantifying land-atmosphere exchanges of water, heat, and CO2 is vital for understanding the Earth system responses to climate change. The rationale of these study has also been revised. Check the contents in line 84-96, 107-108 of the revised manuscript.
- (2) Research gaps We emphasize the lack of long-term, spatially distributed eddy-covariance observations over the western and northern TP, where ecosystem types remain underrepresented in global flux networks. This gap limits understanding of how hydrothermal gradients and permafrost conditions regulate surface fluxes. Check line 152-155 of the revised manuscript.
- (3) Dataset contribution and application The revised text clarifies that the newly established multi-site observation network provides the coordinated, standardized measurements of energy, water, and CO2 fluxes across diverse TP landscapes. This dataset enables cross-ecosystem comparisons and process-level analyses to improve model parameterizations and remote-sensing validations over high-elevation regions. We now explicitly state that this dataset will support climate model evaluation, flux upscaling, and regional carbon budget assessments, thereby filling an essential observational gap in one of the world's most climate-sensitive regions. Chec line 155-157, line 168-170 of the revised manuscript.

These additions make the "Introduction" more focused, highlighting what new knowledge this dataset enables and its scientific value to the broader research community.

2. The inconsistency in result presentation makes the structure of the paper not very straightforward. For instance, the analysis between energy fluxes and climate variables across dry and wet stations provides interesting results. However, it might be beneficial to clarify why this analysis is particularly necessary for energy fluxes instead of carbon fluxes, and why different temporal resolutions (diurnal, daily, monthly) are shown for different variables as main figures.

**Responses:**

We thank the reviewer for the constructive comments and valuable suggestions.

First, the analysis between energy fluxes and climate variables was emphasized because energy fluxes (sensible and latent heat) respond directly and instantaneously to meteorological drivers such as radiation, air temperature, and humidity. Examining these relationships helps to reveal the physical mechanisms of land – atmosphere coupling across wet and dry stations.

Second, the carbon fluxes (NEE, GPP, and Re) are biologically regulated and generally respond to environmental conditions in a more integrated and lagged manner. Therefore, they are discussed separately in Section 3.4, focusing on seasonal patterns that better represent vegetation phenology and ecosystem carbon dynamics rather than short-term variability. Previous studies (e.g., Wang et al., 2021) have already analyzed the dominant climatic drivers of carbon fluxes over grasslands (10 sites) in the Tibetan Plateau. Our newly established western sites, in contrast, exhibit relatively weak NEE variations, and here we primarily present the diurnal and seasonal characteristics of NEE, GPP, and Re to illustrate the dataset 's representativeness, as shown in Figure 6 and Figure S6.

The aim of this study is to introduce and validate a new multi-station flux dataset for the Tibetan Plateau. The analyses included in this paper are intended mainly to demonstrate data quality and highlight notable physical and ecological patterns that can support future in-depth research. To further improve clarity, we have strengthened the transitions between subsections to better link meteorological, hydrological, and carbon processes. We have also added a brief summary paragraph at the end of each subsection. These revisions help clarify the focus of each section. Check line 368-372, line 416-420, 520-524, 596-601 of the revised manuscript.

**References:**

Wang, Y., J. Xiao, Y. Ma, Y. Luo, Z. Hu, F. Li, Y. Li, L. Gu, Z. Li, and L. Yuan (2021), Carbon fluxes and environmental controls across different alpine grassland types on the Tibetan Plateau, Agricultural and Forest Meteorology, 311, 108694

**Specific comments:**

1. L108-116: These results are very specific, even presented with numbers. Are these results specific to TP? There are many land surface models calibrated with the EC flux measurements. Is there a particular reason to highlight this PML-V2 model here? And why is the predictability of ET particularly introduced here, instead of, for instance, GPP?

**Responses:**

We thank the reviewer for this insightful comment. The results mentioned in this section are indeed specific to the Tibetan Plateau (TP), as both the PML-V2 model and the improved ET model by Yuan et al. were validated using eddy covariance (EC) observations from TP sites. These studies are highlighted because they report consistent ET estimates (≈350 mm yr¹) across independent datasets and demonstrate robust model performance in representing ET over the TP, thereby providing reliable references.

The PML-V2 model is particularly emphasized because it explicitly couples water and carbon fluxes and has been widely applied for regional-scale ET estimation, including over the TP. Nevertheless, we acknowledge that EC validation sites in the western TP remain sparse, and our newly established stations offer valuable additional data to improve future model calibration and spatial representation in this under-observed region.

We would like to clarify that our intention was not to focus solely on ET, but rather to highlight its relevance as part of the broader water—energy—carbon coupling central to land—atmosphere (LA) interactions on the TP. The importance of the dataset for carbon fluxes (GPP, NEE, and Re) is also emphasized in following contents. We have revised the relevant sentences (Lines 135–138) in the manuscript to clarify these points and prevent potential misunderstanding.

2. L128: In addition to the spatial patterns, temporal variations can also be highlighted.

**Responses:**

We thank the reviewer for this insightful suggestion. We have revised the sentence following the comments, check line 150 of the revised manuscript.

3. L142-152: These sentences seem to belong to the introduction section.

**Responses:**

We thank the reviewer for this helpful comment. We agree that the content provides general background on EC systems and global flux networks. While, we intentionally placed these sentences at the beginning of section 2.1 to serve as a transition between the introduction and the data description. This paragraph links the global context of eddy covariance networks to our specific observational network on the Tibetan Plateau (TP), introducing why such measurements are essential under the TP's unique environmental conditions.

To make this intent clearer, we have slightly revised the text to emphasize its role as a contextual bridge leading into the methodology section. Check line 173-179 of the revised manuscript.

4. L172: Are the turbulent fluxes measured on the same 20 m PBL tower? How would the tall tower affect the EC measurements?

**Responses:**

We thank the reviewer for this insightful question. All the turbulent fluxes were measured using the EC system mounted on the 20 m PBL tower. At each site, the

main wind direction and surrounding terrain conditions were carefully considered when installing the sensors to avoid flow distortion and ensure representative flux footprints. In addition, all data underwent standard quality-control and turbulence tests (e.g., spike detection, stationarity, and integral turbulence characteristics), confirming that the tower setup did not introduce enormous bias in the EC measurements.

This clarification has been added to Lines 202-204 of the revised manuscript.

5. L179: Does the "CO2 exchange" have the same meaning as "carbon source/sink dynamics"?

**Responses:**

We thank the reviewer for pointing out this important clarification. In our study, "CO2 exchange" refers to the measured bidirectional flux of CO2 between the land surface and the atmosphere, as obtained from the EC system. In contrast, "carbon source/sink dynamics" describe the temporal patterns and direction of this exchange — that is, whether the ecosystem acts as a net carbon sink (CO2 uptake) or a net carbon source (CO2 release) during specific periods. We have revised the sentence to avoid misunderstanding. Check line 206-208 of the revised manuscript.

6. What are the precise vegetation types? What is the forest species at the Medog site? It's helpful to provide detailed information for further research.

**Responses:**

We thank the reviewer for this helpful comment. We agree that detailed vegetation information can be valuable for ecological and land surface studies. However, since this study primarily focuses on atmospheric processes and land – atmosphere fluxes, the available field records and observation network mainly classify land surface types at the ecosystem level (e.g., alpine meadow, grassland, forest, or barren land) rather than by detailed vegetation species. These information has been included in Table 1.

For the Medog site, the underlying surface is characterized as subtropical evergreen broadleaf forest, i.e. banana trees, but detailed species composition data are not yet available due to the complex vegetation species. We have added these information in the revised manuscript. Check line 232-233 of the revised manuscript.

7. Sometimes, it is difficult to capture information from the text descriptions. For instance, the paragraph L184-196 can be more informative if summarized into a table.

**Responses:**

We appreciate the reviewer's helpful suggestion. The basic information for all observation stations, including their geographic location, land cover type, elevation, and instrumentation, is already summarized in Table 1. The corresponding text in Lines 184–196 provides supplementary descriptive context, such as site surroundings, vegetation characteristics, and system upgrade history, to help readers better understand the environmental settings of each station.

Therefore, we retained the textual descriptions to complement Table 1, while ensuring the paragraph remains concise and consistent with the tabulated information.

8. L239-242: Are most of the data gaps related to precipitation?

**Responses:**

We thank the reviewer for this thoughtful question. In fact, the data gaps are not directly related to precipitation events. The missing records mainly resulted from instrumental failures, power interruptions, and occasional human operational errors during maintenance or data collection. These technical issues caused intermittent losses at a few stations (QOMS, NAMORS, and SETORS), as detailed in Table S1 of the manuscript.

To ensure comparability among stations, we analyzed precipitation data covering a full annual cycle (July 2021–June 2022), during which the data loss was 20 % at QOMS, 22 % at NAMORS, and 6 % at SETORS. The rain gauge (RG3, Onset) records only liquid precipitation during warm seasons and does not capture solid precipitation in cold months. Therefore, the missing data—mainly from winter and pre-monsoon periods—do not affect the conclusions regarding the variations in liquid precipitation discussed in this study.

These contents have been revised in line 267-275 of the revised manuscript to avoid misunderstanding.

9. How to define "abnormal values" (L246)? What do you mean by "carefully checked and removed" (L249)? And how is the gap-filling performed (L261)? Please provide clear data processing guidelines.

**Responses:**

We thank the reviewer for the helpful comments. In the revised manuscript, we have provided detailed definitions and methodological explanations for identifying abnormal values, data screening, and gap-filling.

Abnormal values refer to physically implausible or instrumentally corrupted data points identified through range tests, consistency checks, and spike detection following standard FLUXNET and REddyProc protocols. For example, downward

shortwave radiation greater than the solar constant (~1360 W m-2) or less than 0 W m-2, and meteorological variables showing unrealistic diurnal or seasonal patterns due to power or sensor failures, were flagged and removed.

"Carefully checked and removed" means that flagged data were visually inspected and cross-checked with auxiliary records (e.g., diagnostic flags and sensor status) to ensure that only data affected by clear malfunctions or interference were excluded.

Gap-filling was conducted using the marginal distribution sampling method implemented in REddyProc [Wutzler et al., 2018]. This algorithm fills missing NEE data based on relationships with radiation, temperature, and vapor pressure deficit within a 7–14 day moving window, ensuring physically consistent estimates.

These clarifications have been incorporated into the revised manuscript in Lines 276-280, 294-301.

10. Why is NEE calculated, not measured?

**Responses:**

We thank the reviewer for this important clarification. We agree that our previous description was imprecise. In EC measurements, NEE is directly measured as the net vertical flux of CO2 between the ecosystem and the atmosphere, while GPP and Ecosystem Respiration are calculated from NEE through flux partitioning.

We have revised the text accordingly to accurately describe that NEE is measured using the EC system, and that GPP and Re are derived variables estimated by the REddyProc package following standard partitioning procedures. Check line 287-290 in the revised manuscript.

11. L325: Isn't precipitation also one of the meteorological variables in section 3.1?

**Responses:**

We thank the reviewer for this insightful comment. We agree that precipitation is indeed a key meteorological variable. While, in this study we intentionally discuss precipitation and soil water content together in Section 3.2 because both variables are closely related to hydrological dynamics, which are analyzed jointly to illustrate water availability and its impacts on surface fluxes. In contrast, Section 3.1 focuses mainly on atmospheric meteorological conditions—such as air temperature, air humidity and wind speed—that primarily describe energy and atmospheric states rather than water storage or flux processes.

This structure helps maintain thematic clarity between atmospheric variables (Section 3.1) and hydrological variables (Section 3.2), and we consider to keep the original structure in the revised manuscript.

12. Figure 4: why are the data aggregated at monthly values? It can be informative to show the data for each day of year as well (e.g., Figure 2 in Paulus et al., 2022)

**Responses:**

We appreciate the reviewer's helpful suggestion. The data were aggregated to monthly values primarily to highlight seasonal trends and to reduce short-term variability caused by synoptic weather events and measurement noise, which can obscure broader temporal patterns. Monthly aggregation allows for clearer visualization and interpretation of seasonal cycles and cross-site comparisons.

However, we agree that daily-scale variations can provide additional insight into short-term ecosystem responses. The monthly averaged diurnal variation of SH and LE can be found in Figure S4 and Figure S5 respectively, all these results suggest that the published turbulent heat flux show reasonable spatial distribution and temporal variations over the Tibetan Plateau. And we believe that the published data set can be widely accessible to researchers for more detailed analysis in the future.

13. L409-414: Can you give info about which reason(s) applies for which site(s)?

**Responses:**

We appreciate the reviewer's thoughtful questions. We agree that identifying which factors specifically contribute to energy balance closure ratio (EBR) at each site would enhance understanding. However, as the observed EBR variability arises from the combined influence of instrumental setup, surface heterogeneity, and local meteorological conditions, it is challenging to quantitatively separate the contribution of each factor without a dedicated footprint or energy storage analysis, which is beyond the scope of the present study.

In the revised text, we have clarified that the potential reasons listed (e.g., footprint mismatch, advection, canopy storage, high/low frequency losses) may act simultaneously to varying degrees across different sites, depending on their terrain and climatic context. We have also added a sentence noting that future work will focus on site-specific assessments of energy balance closure and its controlling factors. Check the contents in line 462-467 of the revised manuscript.

14. L435: How is DT and DE calculated? Why is DT analysed only with SH instead of LE?

**Responses:**

We thank the reviewer for this insightful question. In the revised manuscript, we have clarified the calculation of DT and DE, as well as the rationale for analysing DT only in relation to sensible heat flux (SH) rather than latent heat flux (LE).

Firstly, DT represents the temperature gradient between the land surface and the atmosphere, calculated as the difference between land surface temperature and air temperature. DE represents the vapor pressure deficit representing the difference between the saturation vapor pressure and the actual vapor pressure of the air.

Secondly, the sensible heat flux (SH) is primarily driven by temperature gradients (DT) and turbulent exchange, hence DT – SH relationships reflect the efficiency of heat transfer. The latent heat flux (LE), on the other hand, depends on vapor pressure deficit (DE) rather than temperature gradients. Therefore, LE is analysed with DE, not DT, to correctly represent the underlying physical process of moisture transfer.

These clarifications have been added in the revised manuscript to enhance the clarity of the parameter definitions and their physical relevance. Check line 468-470 of the revised manuscript.

**15. Figure 5: Are these correlation coefficients all statistically significant?**

**Responses:**

We thank the reviewer for this important question. We have added statistical significance for all reported correlation coefficients using Pearson's correlation analysis (two-tailed test) throughout the manuscript, and the corresponding p-values have now been added in the revised manuscript and are indicated either in parentheses or using asterisks in figures and tables (p < 0.05 or p < 0.01), i.e. Table S7, Figure S1-S2.

The averaged correlation coefficients from Figure 5 all come from the values from Table S7, where the majority of the correlation coefficients pass the statistical significance test (p < 0.05).

These information has been added in the revised manuscript (Check Table S7, Figure S1-S2 and many others) and this addition improves transparency and supports the validity of the reported relationships.

16. L456: Why is it doubled? This sentence is misleading.

**Responses:**

We appreciate the reviewer's observation. We agree that the original wording may give the misleading impression that the relationship between temperature and saturation moisture content is linear. In fact, the saturation vapor pressure (and hence the air's capacity to hold moisture) increases exponentially with temperature, at an approximate rate of 6–7% per °C according to the Clausius–Clapeyron relationship.

We have revised the sentence to accurately describe this behavior:  $^{\circ}\Delta E$  represents the difference between the actual and saturation vapor pressure, indicating the atmospheric moisture deficit. The saturation vapor pressure increases exponentially with temperature — by about 6-7% per degree Celsius — so warmer air can hold substantially more water vapor than cooler air."

This correction removes the misleading "doubled" statement and improves scientific accuracy. The contents can be found in Lines 511-514 of the revised manuscript.

17. L526-530: It is not clear what exactly leads the site Medog to be a carbon source. Is it contributed by disturbances or climate extremes? What causes the strong seasonal variation of GPP at this site (Figure 6)?

**Responses:**

We thank the reviewer for this valuable comment. In the revised manuscript, we have clarified that the carbon source pattern at Medog is primarily attributed to recent site disturbances associated with tower construction, which led to partial vegetation removal and soil exposure. These disturbances enhanced soil and microbial respiration, contributing to net CO2 release. Additionally, the complex topography in the region causes uneven solar radiation exposure and microclimatic variation, which, combined with a hot and humid monsoonal climate, influence photosynthetic activity.

The strong seasonal variation in GPP (also for NEE and Res) in Figure 6 mainly reflects the monsoon-driven fluctuations in radiation and temperature, with GPP peaking during the warm, wet season and decreasing under cooler, cloudier conditions. As NEE measurements are not complete throughout the whole year, we have also removed the filled data of GPP and Res by REddyProc for consistency (Figure 61). Thus, these contents have all been revised in the revised manuscript. The explanation can be found in Lines 575-585 for clarity.

**Technical comments:**

1. L137: Incomplete sentence: "...are introduced..."

**Responses:**

Thanks for the correction. We have revised the whole sentence to avoid the wrong grammar issue. Check line 160-162 of the revised manuscript.

2. L229: analyze -> analyzing;

**Responses:**

Thanks for the corrections. We have revised the word following the comment. Check line 258 of the revised version.

3. L231: analyzing -> analyze

**Responses:**

Thanks for the corrections. We have revised the word following the comment. Check line 260 in the revised version.

4. L369: Do you mean energy flux?

**Responses:**

Thanks for the comments. We agree that "energy flux" is more accurate term. The phrase "energy budget flux" has been corrected to "energy flux". Check the revised content in line 422.

References: Paulus, S. J., El-Madany, T. S., Orth, R., Hildebrandt, A., Wutzler, T., Carrara, A., Moreno, G., Perez-Priego, O., Kolle, O., Reichstein, M., and Migliavacca, M.: Resolving seasonal and diel dynamics of non-rainfall water inputs in a Mediterranean ecosystem using lysimeters, Hydrol. Earth Syst. Sci., 26, 6263–6287, https://doi.org/10.5194/hess-26-6263-2022, 2022.